Transcriptome-based analysis of the hormone regulation mechanism of gender differentiation in Juglans mandshurica Maxim

Qin Baiting 1 2
Lu Xiujun 1
Sun Xiaomei 1
Cui Jianguo 1 2
http://orcid.org/0000-0001-5496-5009 Deng Jifeng 1 2
Zhang Lijie 1 2 Zlj330@syau.edu.cn
1 College of Forestry, Shenyang Agricultural University , Shenyang , China
2 Key Laboratory of Forest Tree Genetics, Breeding and Cultivation of Liaoning Province , Shenyang , China
Fukushima Atsushi
Electronic publication date: 2021 Nov 9
Publication date: 2021
Volume: 9
Electronic Location ID: e12328
Received 2021 Jan 26; Accepted 2021 Sep 27
Copyright: © 2021 Qin et al.
Copyright year: 2021
Copyright holder: Qin et al.
License: This is an open access article distributed under the terms of the Creative Commons Attribution License, which permits unrestricted use, distribution, reproduction and adaptation in any medium and for any purpose provided that it is properly attributed. For attribution, the original author(s), title, publication source (PeerJ) and either DOI or URL of the article must be cited.
License URL: https://creativecommons.org/licenses/by/4.0/

Keywords: Juglans mandshurica maxim, Heterodichogamy, Transcriptome sequencing, Hormonal regulation

Funding: Liaoning Education Department LSNZD201905 Liaoning Natural Science Foundation of China 20170540801 13th 5-year National Key Research and Development Plan Project 2017YFD0600600 This work was supported by the Scientific Research Project of Liaoning Education Department (LSNZD201905), the Key Project of Liaoning Natural Science Foundation of China (No. 20170540801) and the 13th 5-year National Key Research and Development Plan Project (No. 2017YFD0600600). The funders had no role in study design, data collection and analysis, decision to publish, or preparation of the manuscript.

==============================
Juglans mandshurica Maxim is a hermaphroditic plant belonging to the genus Juglans in the family Juglandaceae. The pollination period of female flowers is different from the loose powder period of male flowers on the same tree. In several trees, female flowers bloom first, whereas in others, male flowers bloom first. In this study, male and female flower buds of J. mandshurica at the physiological differentiation stage were used. Illumina-based transcriptome sequencing was performed, and the quality of the sequencing results was evaluated and analyzed. A total of 138,138 unigenes with an average length of 788 bp were obtained. There were 8,116 differentially expressed genes (DEGs); 2,840 genes were upregulated, and 5,276 genes were downregulated. The DEGs were classified by Gene Ontology and analyzed by Kyoto Encyclopedia of Genes and Genomes. The signal transduction factors involved in phytohormone synthesis were selected. The results displayed that ARF and SAUR were expressed differently in the auxin signaling pathway. Additionally, DELLA protein (a negative regulator of gibberellin), the cytokinin synthesis pathway, and A-ARR were downregulated. On April 2nd, the contents of IAA, GA, CTK, ETH and SA in male and female flower buds of two types of J. mandshurica were opposite, and there were obvious genes regulating gender differentiation. Overall, we found that the sex differentiation of J. mandshurica was related to various hormone signal transduction pathways, and hormone signal transduction plays a leading role in regulation.

Introduction

Heterosexualism is a transitional type in the evolution of plants from monoecious to dioecious. In the evolution of flowering plants from monoecious to dioecious, the evolution of the reproductive system has been differentiated by male and female flowers to form two mating types, namely, female and male flowers (Gleuser et al., 2008; Teichert, Dötterl & Gottsberger, 2011; Wang et al., 2012; Fukuhara & Tokumaru, 2014; Renner, 2014; Sun & Verdu, 2016). The dioecious mating system is not common in flowering plants (Wang, 2001). At present, only Batulaceae, Corylus, Juglandaceae, Carya, Juglans, and Cyclocarya are woody plants with a monoecious mating system. Other Acer species of Aceraceae belong to bisexual flowers, but they are unisexual or monoecious in function. There are many patterns of flower expression sequences in dioecious plants, but generally speaking, these two mating types are fixed (Fu et al., 2010; Tal, 2009), and in most species of Juglans, the ratio of males to females in the mating-type is 1:1.

Sex-determining genes are the genetic basis of sex differentiation, and hormones play an important role in regulating flower sex differentiation as induction signals in the process of plant sex differentiation (Golenberg & West, 2013; Aryal & Ming, 2014; Aryal et al., 2014; Kong, 2009). These endogenous hormones regulate flower induction and formation through various flowering pathways (photoperiod pathway, gibberellin pathway, spring flower pathway, and autonomous pathway) (Mutasa-Göttgens & Hedden, 2009; Amasino, 2010; Lee & Lee, 2010; Xing et al., 2015, 2016; Shimakawa et al., 2012). Molecular studies have revealed that plant hormones are one of the induction signals of plant gender differentiation and play an important role in regulating the process of gender differentiation (Tanurdzic & Banks, 2004; Kong, 2009). However, there is a complex interaction among the hormones that makes the regulation mechanism of plant hormones on gender determination and differentiation extremely complex, and there is no unified regulation model (Li, 2016; Yu et al., 2020; Zhong & Qin, 2010). The same hormone may lead to the expression of different gender differentiation programs in different plants through the difference in the signal transduction process (Xing, 2007; Wang & Zeng, 1996; Wang & Zeng, 1997; Chen et al., 2005; Anusak et al., 2003; Arora et al., 2007). To date, the control theory of hormone-regulating sex determination has focused on several main economic plants but has not been reported in dioecious woody plants. Research on the flowering mechanism of dioecious woody plants is limited to the basic research on cytohistology and molecular biology of male and female flower bud differentiation.

Juglans mandshurica is a broad-leaved tree species of Juglans in Juglandaceae. It is also an important woody tree species for grain and oil and an economic tree species for both fruit and wood in China (Zhang et al., 2019b). Juglans mandshurica fruit and wood have broad prospects for development and utilization. In recent years, a research group collected and investigated the germplasm resources of J. mandshurica in Northeast China and found that J. mandshurica has monoecious heteromorphism, and there are mainly two mating types: protandrous and protogynous. The male inflorescence of protandrous is synchronized with the spreading leaf, and the female flower begins to bud after the male flower matures and scatters; the development of female flowers of protogynous is synchronous with leaf spreading, while the development of male flowers is slow. After the pollination of female flowers is complete, the male flowers grow rapidly and mature and scatter pollen. The fertile period of female flowers is different from the scattering period of male flowers. The flowering sequence of males and females is very stable, and the two mating types are randomly distributed in the population (Zhang et al., 2019a). This reproductive characteristic leads to a lack of flowering time between male and female flowers, which seriously affects the pollination rate and fruit setting rate. Therefore, in this study, the molecular mechanism of hormone regulation in the process of sexual differentiation of J. mandshurica was analyzed using high-throughput Illumina transcriptome sequencing technology, and the interaction between sexual differentiation and endogenous hormone synthesis and signal transduction was explored. The results of this study can be used to guide forestry production, and provide an effective theoretical basis and technical support for improving the seed yield and quality of J. mandshurica by rationally allocating the proportion of different mating types of plantation populations and manually controlling pollination.

Materials & methods

Plant material

Based on the previous research results of the phenology, morphology, physiology, and cytology of J. mandshurica natural forest in eastern Liaoning, China, the morphological development period, physiological differentiation period, and vegetative growth period of J. mandshurica flower bud differentiation process were determined. In 2019, the flower buds were selected on March 17 and 25, and April 2, 10, and 15 to be quick-frozen in liquid nitrogen and stored in a refrigerator at –80 °C for differential gene expression analysis and metabolomics. Simultaneously, the female and male flower buds collected on April 2 during the physiological differentiation period of the male and female precursors (Fig. 1) were selected for transcriptomic research. The female flower buds of the female precursor type were named T1, which were repeated three times, namely A1, A2, and A3, respectively. The male flower buds were named T2, which were repeated three times, namely A4, A5, and A6. The female flower buds of the male precursor type were named M1, which were repeated three times, namely B1, B2, and B3. The male flower buds were named M2, which were repeated three times, namely B4, B5, and B6. There were 12 samples in total.

Figure 1 (A) Male and (B) female flower buds of Juglans mandshurica.

Construction and sequencing of library

Oligo(dT) magnetic beads were used to enrich mRNA with polyA structure in total RNA, and the RNA was interrupted to 200–300 bp fragments by ion interruption. After the construction of the library was completed, PCR amplification was adopted to enrich the library fragments, and then library selection was carried out according to the fragment size, where the library size is 300–400 bp. Next, the library was inspected using an Agilent 2100 Bioanalyzer, and the total concentration and effective concentration of the library were detected. After RNA extraction, purification, and library building, these libraries were sequenced using the second next-generation sequencing technology based on an Illumina HiSeq sequencing platform. The sequencing raw read length was 150 bp, and the original sequence of mRNA in the physiological differentiation stage of male and female flower buds of J. mandshurica was obtained.

Sequencing data sorting and transcript assembly

We counted the Raw Data of each sample separately, spliced Clean Reads with Trinity software to obtain transcripts, and then carried out the subsequent analysis. Trinity was used for D. novo assembly software for transcriptome splicing, which splices high-quality sequences based on the De Bruijn Graph splicing principle. After splicing was completed, the transcript sequence file in the FASTA format was obtained. The longest transcript under each gene was extracted as the representative sequence of the gene, called unigene.

Unigene annotation classification and expression analysis

The database used for gene function annotation included NR (NCBI non-redundant protein sequences), GO (Gene Ontology), KEGG (Kyoto Encyclopedia of Genes and Genome), eggnog (evolutionary genealogy of genes: Non-supervised Orthologous Groups), and Swiss-Prot.

Differential expression and enrichment analysis

By comparing the RNA-seq data of the physiological differentiation stage of female and male flower buds, the gene expression was analyzed by DESeq. The conditions for screening differentially expressed genes were as follows: expression difference multiple |log2FoldChange| > 1, significant P-value < 0.05. Significant enrichment analyses were conducted of differentiated genes based on the number of differentially expressed genes contained at different levels of each KEGG pathway, after multiple tests and corrections, the pathway with a Q-value ≤ 0.05 was selected as the pathway significantly enriched in differentially expressed genes, and then the metabolic pathway and signal pathway in which differentially expressed genes were mainly involved were determined.

qRT-PCR verifies sequencing data

Twelve differential expression genes were randomly selected, and their expression profiles were detected via qRT-PCR to verify transcriptome data.

Endogenous hormone determination

Samples from different periods were tested for phytohormones, including IAA, GA, CTK, ABA, ETH, BR, JA, and SA, then analyzed and compared (Weiler, Jourdan & Conrad, 1981).

Results

Transcriptome sequencing analysis of male and female flower buds

Raw transcriptome sequencing reads all flower bud samples with raw sequencing reads of more than 40 M, an average reading length of 150 bp, a total number of sequencing bases of over 6G nt, a Q20 content of over 95%, and a Q30 content of over 90%. Statistical analysis revealed that the sequencing data were of high quality and could meet the requirements of further analysis. The data filtering used Cutadapt to remove the 3′-end linker, and the removed part had at least 10 bp overlap (AGATCGGAAG) with the known linker, allowing a 20% base mismatch. Remove read with an average mass fraction lower than Q20. The number of clean reads obtained from female flower buds of female precursor type ranged from 43,894,726 to 46,469,762. The number of high-quality sequence reads was 6,610,627,884 to 6,991,929,010, the percentage of high-quality sequence reads in sequencing reads was 99.41–99.49%, and the percentage of high-quality sequence bases in sequencing bases was 99.13–99.16% (Table S1).

Using the reads assembly software Trinity to assemble from scratch, the assembly fragments contig were obtained according to the overlap of reads, and these contigs were linked together by overlap to obtain unigene. Transcript and unigene sequences were 344,218 and 138,138, respectively, and the transcript and unigene sequences had average lengths of 1,182 bp and 788 bp, respectively. The number of unigene in the physiological differentiation stage of male and female flower buds of J. mandshurica was approximately 80,000, while that of unigene larger than 500 bp was almost 60% of the total. This indicated that the sequencing results were of high splicing quality, which was conducive to further functional annotation analysis (Fig. 2).

Figure 2 Length distribution of unigene for transcriptome in J. mandshurica.

All-unigene obtained by transcriptome sequencing was annotated with protein and nucleic acid functions (Table 1). Among the 108,828,259 bp unigene assembled in total, all-unigene sequences were sequentially aligned to the protein database through blastx, and unigene was annotated with gene functions. The databases used for gene function annotation included NR (NCBI Non-Redundant Proteins Sequences), GO (Gene Ontology), KEGG (Kyoto Encyclopedia of Genes and Genome), eggNOG (evolutionary genealogy of genes: Non-supervised Orthologous Groups), and Swiss-Prot. The number of successful annotations of unigene into the Nr protein database reached 60,415, accounting for 43.74% of the total number of all-unigene. The number of unigenes annotated in the GO database reached 29,402, accounting for 21.28% of the total unigenes. The number of unigenes annotated in the eggNOG database reached 57,250, accounting for 41.44% of the total unigenes. The number of unigenes annotated in the Swissprot database reached 45,347, accounting for 32.83% of the total unigenes, while the number of unigenes annotated in the KEGG database was at least 6,845, accounting for only 4.96% of the total. The number of unigenes annotated in all databases was 5,312, accounting for only 3.85% of the total unigenes.

Table 1 Notes on unigene function of Juglans mandshurica transcriptome.

Database	NR	GO	KEGG	EggNOG	Swissprot	In all database	
Number	60,415	29,402	6,845	57,250	45,347	5,312	
Percentage (%)	43.74	21.28	4.96	41.44	32.83	3.85	

According to the statistics of unigenes annotated in the Nr protein database by E-value distribution, e-values in 1e−5–1e−15 accounted for 26.72%, and 1e−15–1e−30 accounted for 22.33% (Fig. S1A). According to the annotation similarity distribution statistics, 19.45% of the sequences were concentrated in the 80–90% range, with a total of 55.51% sequence similarity higher than 60% (Fig. S1B). Homology analysis among different species demonstrated that Vitis vinifera had the highest homology, reaching 19.26%, followed by Ziziphus jujuba (6.29%), and other species (58.27%). The distribution of species homology revealed high homology with grape (Fig. S1C).

The gene was annotated by eggNOG comparison, and the eggNOG number of the best comparison result was assigned to the corresponding gene. Further, using the correspondence between the eggNOG number and the eggNOG classification catalog, each gene was classified into the eggNOG classification catalog. A total of 77,173 unigenes were compared to the eggNOG database (Fig. 3). Among them, 21,187 were compared with the predicted general function prediction only, and 15,111 were compared with the predicted unknown function. Then, compared with the predicted number of aggregation classifications, the sequence was 4,728 signal transmission mechanisms, 4,354 transcriptions, 4,069 replications, recombination, and repair. Post-translational modification, protein transformation, 4,042 molecular chaperones, 2,896 translations, ribosome structure, and biogenesis. The above approach demonstrated that substance and energy metabolism were very vigorous in the process of J. mandshurica flower bud differentiation, cell growth, proliferation, and division speed were very fast, and hormone signal transduction participated in the differentiation process.

Figure 3 EggNOG function classification of unigene for transcriptome in J. mandshurica.

GO annotation was completed using BLAST2 GO software, and the annotation adopted the default parameters of BLAST2 GO. The results of the GO annotation were mapped to go term, and the number of genes annotated in the second-level classification was counted (Fig. S2). According to classification statistics, there were 29,402 (21.8%) unigene genes annotated in the J. mandshurica transcription group into the GO database. There were three major categories of annotations: 32,959 (24.5%) of molecular function, 51,672 (38.4%) of biological processes, and 50,032 (37.2%) cellular components. There were 67 functional groups. Among the 24 functional groups of biological processes, unigene included 14,613 metabolic processes (GO:0008152) and 14,514 cellular processes (GO:0009987). There were 10,152 cells (GO:005623) and 10,063 cells part (GO:0044464). Among the 19 functional groups of molecular functions, there were 14,843 bindings (GO:0005488) and 14,061 catalytic activities (GO:0003824). Through GO function annotation, a large number of unigenes related to metabolic process, biological regulation, cell process, cell, membrane, and membrane component organelles, and other unigenes were obtained in the three major types of functions. These genes can provide a theoretical basis for the study of sex differentiation and flowering related functional genes of J. mandshurica.

A total of 6,288 unigenes were mapped to 35 KEGG metabolic pathways, of which 606 genes were annotated by the signal transduction pathway in environmental information processing. A total of 504 genes were annotated by the translation pathway in genetic information processing, and 487 genes were annotated by carbohydrate metabolism in the metabolic pathway (Fig. S3).

Analysis of GO and KEGG pathway in male and female flower buds

According to the classification and annotation of the GO function of differentially expressed genes in male and female flower buds of female precursor type (T1 vs T2), we can observe that there were 837 functions, including 230 molecular functions, 517 biological processes, and 90 cell compositions. We analyzed the significance of GO function enrichment of female and male flower buds of female precursor type (when Bonfeeoni-corrected P-value ≤ 0.05). The results revealed that T1 vs. T2 significantly enriched 36 terms, 18 terms belonged to the biological process, eight terms belonged to molecular function, and 10 terms belonged to cellular component (Fig. 4). The enriched term included nucleic acid binding transcription factor activity (GO:0001071), regulation of RNA biosynthesis processes (GO:2001141), ribosomal subunit (GO:0044391), etc. The cluster frequencies are 6.1%, 6.7%, and 9.9%, respectively. The main annotated genes include GAI, SOC1, AP1, etc. The metabolism of endogenous substances changed during the physiological differentiation of female and male flower buds of J. mandshurica, and, simultaneously, the genes related to flower formation were pre-activated to a certain extent, laying a foundation for the differentiation of flower organs. There were 121 genes annotated on the pathway compared in T1 vs. T2. Differential Expression Genes (DEGs) significantly enriched 20 pathways in KEGG (Fig. 5), of which plant hormone signal transduction was the highest (ko04075). The second was carbon metabolism (ko01200). Additionally, glyoxylate and dicarboxylate metabolism (ko00630), AMPK signaling pathway (ko04152), and zeatin biosynthesis (ko00908) were also significantly enriched. These results demonstrate that hormones play a leading role in regulating the differentiation of female and male flower buds of the female precursor type, and carbohydrate was also an essential metabolic substance during the transformation from physiological differentiation of flower buds to morphological differentiation.

Figure 4 GO annotation analysis between male and female flower buds of female precursor Juglans mandshurica (T1 vs. T2).

Figure 5 KEGG pathway enrichment of female and male flower buds of female precursor J. mandshurica (T1 vs. T2).

Male and female flower buds of male precursor type (M1 vs. M2) had a total of 1,013 functions annotated, including 649 biological processes, 94 cell compositions, and 270 molecular functions (Fig. 6). A total of 54 terms were significantly enriched, included 28 belonged to biological process, 21 belonged to molecular function and five belonged to cellular components. The enriched term included protein dimerization activity (GO:0046983), regulation of transcription, DNA-templated (GO:0006355), extracellular region (GO:0005576), etc. The cluster frequencies are 5.6%, 9.8%, and 4.0%, respectively. The main annotated genes include ARF5, GA2OX1, TEM1, SOC1, etc. A total of 125 differentially expressed genes DEGs of M1 vs. M2 were enriched in the KEGG pathway (Fig. 7). Among them, the ribosome (ko03010) was the most abundant, followed by plant hormone signal transmission (ko04075) and carbon metabolism (ko01200), which indicated that M1 vs. M2 was affected by gene regulation, hormone regulation, and physiological metabolites in the process of sex differentiation. The KEGG significant enrichment analysis (Bonferroni corrected P-value ≤ 0.05) revealed that the KEGG of M1 vs. M2 significantly enriched the 18 pathways. Among them, plant hormone signal transduction (ko04075) was the highest, followed by the neurotrophin signaling pathway (ko04722) and starch and sucrose metabolism (ko00500). This indicates that hormones play a leading role in the physiological differentiation of male and female flower buds of male precursor type and regulated the growth and differentiation of organs in all parts of the flower.

Figure 6 GO annotation analysis between male and female flower buds of male precursor Juglans mandshurica (M1 vs. M2).

Figure 7 KEGG pathway enrichment of female and male flower buds of male precursor J. mandshurica (M1 vs. M2).

Analysis of DEGs

According to the screening of differentially expressed genes in the flower bud transcriptome of J. mandshurica, we obtained the DEGs comparison library of female and male flower buds: male pre-type female flower buds and male flower buds (M1 vs. M2, 1,357 genes), male pre-type female flower buds and female pre-type female flower buds (M1 vs. T1, 1,461 genes), male pre-male flower buds and female pre-type male flower buds (M2 vs. T2, 3,476 genes), female pre-type female flower buds, and male flower buds (T1 vs. T2, 1,822 genes) (Fig. 8A). This indicated that most of the DEGs appeared in flower buds of different genders, and 11 of them were differentially expressed in the four libraries. A total of 8,116 differentially expressed genes were obtained in the four cDNA libraries, including 2,840 upregulated genes and 5,276 downregulated genes (total 8,116 genes) in Figs. 8A and 8B. The downregulated genes in male and female flower buds of the two mating types were generally higher than those of the upregulated genes. In M1 vs. M2, 585 genes were upregulated and 772 genes were downregulated. In T1 vs. T2, 359 genes were upregulated and 1,463 genes were downregulated (Fig. 8B) (Data related to differential expressed genes have been submitted to NCBI database and TSA database, TSA SUB10193123).

Figure 8 Number of differentially expressed genes in male and female flower buds (A stands for IAA related differentially expressed transcription factors, B stands for ETH related differentially expressed transcription factors).

In the signal transduction and synthesis pathways of plant hormones, we compared the differentially expressed genes in M1 vs. M2 and T1 vs. T2 libraries and found that the sex differentiation of flower buds was mainly related to ARF, SAUR, DELLA, TF, PR-1, etc (TSA SUB10193123, also uploaded as the Supplementary File, Data related to differential expressed genes. zip).

Differential expression of TFs in male and female flower buds

A total of 195 TFs were differentially expressed in M1 vs. M2, 168 TFs were differentially expressed in T1 vs. T2, and 83 TFs were differentially expressed in M1 vs. M2 and T1 vs. T2. The differentially expressed TFs included members of ARF, AP2, C2H2, ERF, bHLH, MADS, MYB, WRKY, NAC and other TF family members were included. MADS and bHLH families were dominant among the families. We have screened out many transcription factors related to plant hormones, and based on the results, we have identified transcription factors with differential expression in auxin and ethylene signaling pathways (Fig. 9). There were eight identical differentially expressed transcription factors in M1 vs. M2 and T1 vs. T2.

Figure 9 The differential expressed transcription factors between the male and female flower buds.

Verification of gene expression via real-time PCR

To verify the gene expression pattern identified by RNA-Seq data, the transcription levels of 12 randomly selected DEGs were detected by quantitative PCR. We used these 12 genes for qRT-PCR, with three biological replications in each group. The gene expression pattern detected by qRT-PCR was consistent with the RNA sequence data (Fig. 10). The qRT-PCR experiments confirmed that the DEGs obtained from the assembled transcriptome were accurate and the gene expression profile obtained from RNA-Seq data was reliable.

Figure 10 (A–L) Validation of 12 randomly selected differentially expressed genes (DEGs) derived from RNA-seq using qRT-PCR.

Determination of endogenous hormones in flower buds of J. mandshurica

To explore the relationship between the flower bud differentiation of J. mandshurica and plant endogenous hormones, we determined the content of eight plant hormones in flower buds at the early stage of physiological differentiation. The content of plant endogenous hormones affected flowering time. By measuring the plant hormones in the male and female flower buds of J. mandshurica at the early stage of physiological differentiation (Table 2), we found that the contents of auxin (IAA), brassinolide (BR), and salicylic acid (SA) demonstrated an overall upward trend. Gibberellin (GA) content first increased and then decreased, and a peak appeared on approximately April 10th. The content of cytokinin (CTK) fluctuated: up-down-up. The contents of ethylene (ETH) and abscisic acid (ABA) first decreased and then increased. Jasmonic acid (JA) content in flower buds of female precocious J. mandshurica was in a relatively stable state before April 10th and began to increase after this date, while JA content in flower buds of male precocious J. mandshurica revealed an overall upward trend. Interestingly, on April 2nd, the contents of IAA, GA, CTK, ETH, and SA were all found in the flower buds of J. mandshurica: T1 < T2 and M1 > M2. The female flowers of the female pre-type J. mandshurica open first, and the male flowers disperse pollen later. The male flowers of the male pre-type J. mandshurica are scattered first, and the female flowers open later. The contents of the above five hormones in the male and female flower buds of the two types of J. mandshurica were opposite, indicating that there were genes regulating gender differentiation in these five hormones, which was consistent with the differential expression of genes in the transcriptome data. We inferred that these hormones participate in gender differentiation to varying degrees and that there may be interactions. BR and JA contents were as follows: T1 < T2 and M1 < M2. The contents of BR and JA in female flower buds were lower than those in male flower buds, indicating that the regulation of these two hormones was more important in male flower buds. ABA content revealed that: T1 > T2 and M1 > M2, while the ABA content in female flower buds was higher than that in male flower buds, indicating that ABA regulation was more needed in female flower buds.

Table 2 Plant hormone content of J. mandshurica flower buds at different stages.

Hormone	Sample	Date	
3.17	3.25	4.02	4.10	4.15	
IAA
(μg/g)	T1	18.17 ± 1.16ABb	20.79 ± 0.28Aa	25.05 ± 1.21Ab	32.11 ± 2.39Aa	28.78 ± 1.25Bb	
T2	17.14 ± 1.43Bb	16.62 ± 0.76Bb	27.59 ± 0.44Aa	26.32 ± 1.19Bb	39.03 ± 1.31Aa	
M1	21.06 ± 0.88Aa	20.14 ± 0.16Aa	25.23 ± 1.47Ab	31.41 ± 0.85Aa	30.50 ± 1.64Bb	
M2	20.59 ± 0.87Aa	16.55 ± 0.38Bb	24.38 ± 1.03Ab	27.78 ± 1.63ABb	31.43 ± 1.49Bb	
CTK
(ng/g)	T1	151.42 ± 5.40Bb	246.86 ± 14.33Ab	309.37 ± 13.49Aab	260.66 ± 12.97Aab	355.08 ± 11.45Aa	
T2	180.37 ± 4.13Aa	275.90 ± 10.53Aa	320.71 ± 5.52Aa	245.70 ± 12.32Ab	284.82 ± 11.36Bc	
M1	114.23 ± 9.79Cc	163.65 ± 7.30Bc	296.69 ± 10.18Ab	278.77 ± 11.92Aa	347.64 ± 18.50Aa	
M2	126.46 ± 9.83Cc	253.05 ± 11.43Ab	254.93 ± 12.34Bc	266.40 ± 8.15Aab	319.14 ± 14.88ABb	
GA
(μg/g)	T1	0.29 ± 0.005Aa	0.26 ± 0.003Bc	0.31 ± 0.03Aa	0.36 ± 0.03Bc	0.37 ± 0.02Aa	
T2	0.25 ± 0.01Bb	0.32 ± 0.01Aab	0.31 ± 0.02Aa	0.41 ± 0.02ABab	0.33 ± 0.00BCb	
M1	0.20 ± 0.02Cc	0.33 ± 0.02Aa	0.35 ± 0.01Aa	0.38 ± 0.01ABbc	0.31 ± 0.01Cb	
M2	0.22 ± 0.01Cc	0.30 ± 0.02Ab	0.34 ± 0.01Aa	0.44 ± 0.01Aa	0.36 ± 0.01ABa	
ETH
(μg/g)	T1	151.81 ± 10.24Aa	105.50 ± 10.46Bc	104.90 ± 7.35Aa	175.75 ± 9.73Cc	300.46 ± 14.14Aa	
T2	117.94 ± 2.51Bb	119.73 ± 2.78ABb	115.94 ± 8.89Aa	246.95 ± 13.55Aa	273.75 ± 8.71Bb	
M1	139.24 ± 3.33Aa	132.30 ± 5.82Aa	115.89 ± 10.15Aa	193.18 ± 5.82BCbc	232.55 ± 6.02Cc	
M2	147.04 ± 9.30Aa	75.93 ± 2.24Cd	115.00 ± 7.07Aa	211.25 ± 7.83Bb	210.22 ± 6.46Cd	
BR
(μg/g)	T1	0.39 ± 0.02Aa	0.40 ± 0.02BCb	0.49 ± 0.01Bb	0.54 ± 0.04Aa	0.62 ± 0.03Aa	
T2	0.34 ± 0.02Aa	0.35 ± 0.03Cc	0.49 ± 0.03Bb	0.56 ± 0.04Aa	0.60 ± 0.05Aab	
M1	0.39 ± 0.02Aa	0.49 ± 0.02Aa	0.50 ± 0.01Bb	0.55 ± 0.05Aa	0.54 ± 0.04Ab	
M2	0.37 ± 0.04Aa	0.43 ± 0.02Bb	0.56 ± 0.03Aa	0.55 ± 0.04Aa	0.56 ± 0.01Aab	
ABA
(ng/g)	T1	203.23 ± 17.11ABa	129.33 ± 14.13Aa	158.68 ± 7.79Aa	292.41 ± 22.64Bb	377.00 ± 21.01ABa	
T2	101.49 ± 27.99Cc	150.48 ± 20.88Aa	147.02 ± 18.75Aab	217.38 ± 27.47Cc	424.86 ± 34.76Aa	
M1	243.95 ± 16.86Aa	78.98 ± 21.38Bb	154.47 ± 11.68Aa	355.25 ± 7.08Aa	323.42 ± 16.91Bb	
M2	158.68 ± 29.05BCb	70.25 ± 1.60Bb	128.13 ± 13.05Ab	282.78 ± 23.80Bb	399.80 ± 32.65ABa	
SA
(μg/g)	T1	33.18 ± 3.42Bc	46.79 ± 5.34Aa	49.27 ± 2.31Aa	61.62 ± 1.24Aa	65.61 ± 2.10Bb	
T2	48.85 ± 1.91Aa	47.82 ± 0.57Aa	50.62 ± 1.71Aa	55.42 ± 1.99Bb	76.19 ± 2.80Aa	
M1	38.17 ± 1.43Bb	45.12 ± 2.04Aab	47.58 ± 3.17Aa	58.56 ± 1.91ABab	60.57 ± 3.27Bb	
M2	35.32 ± 2.37Bbc	37.55 ± 1.92Ab	46.06 ± 3.65Aa	61.24 ± 2.38Aa	78.22 ± 5.07Aa	
JA
(ng/g)	T1	2.45 ± 0.27ABa	2.40 ± 0.09Ab	2.40 ± 0.21Bb	2.79 ± 0.25Bb	3.83 ± 0.23Ab	
T2	2.54 ± 0.14Aa	2.45 ± 0.08Ab	2.56 ± 0.07Bb	2.68 ± 0.15Bb	4.23 ± 0.21Aa	
M1	2.05 ± 0.07BCb	2.68 ± 0.14Aa	2.95 ± 0.11Aa	3.01 ± 0.02Bb	4.22 ± 0.12Aa	
M2	1.85 ± 0.12Cb	2.70 ± 0.15Aa	3.18 ± 0.06Aa	3.83 ± 0.18Aa	3.35 ± 0.05Bc	

Discussion

Plant endogenous hormones are involved in the regulation of gender differentiation in almost all plants. However, it is generally believed that CTK and GA are the main hormones that affect gender differentiation in plants, while other hormones affect gender expression by changing the activities of GA and CTK (Liu et al., 2016). In this study, according to the sequencing analysis of the flower bud transcriptome of J. mandshurica, it was found that the process of sexual differentiation of J. mandshurica is a complex regulatory network composed of many factors. Furthermore, it is influenced by flowering gene regulation, hormone regulation, and physiological and biochemical metabolites in the process of sexual differentiation. Additionally, hormone played a leading role in the physiological differentiation of male and female flower buds, regulating the growth and differentiation of various organs of flowers. This is consistent with the results of Wu et al. (2010) who found that cucumber sex differentiation genes participate in the signal transmission of plant endogenous hormones by analyzing the differences in transcripts of cucumber plants of different genders. The genes include ACS, ASR1, CSIAA2, CSAUX1, and other genes constituting a regulatory network that works together to regulate cucumber sex differentiation. These plant hormones include GA, ABA, ETH, IAA, CTK, and BR. Additionally, SA, JA, and other plant hormones play key roles in the process of gender differentiation (Conti, 2017; Yamaguchi et al., 2014; Nakagawa et al., 2012; Liron et al., 2014; Papadopoulou et al., 2005; Pan et al., 2014; Chen et al., 2017; Zhu et al., 2015; Khryanin, 2002). The AUX, A-ARR, DELLA, TF, PYL, ERF1/2, CYCD3, and PR-1, which have significant regulatory effects on the flower bud differentiation of J. mandshurica, are classified into IAA, GA, ETH, CTK, ABA, BR, and SA synthesis pathways, and they work together to regulate hermaphroditism. Teo et al. (2014) found that IAA plays an indispensable role in the regulation of inflorescence differentiation in A. thaliana, and the interference of IAA synthesis, transport, and signal transduction leads to pistil or stamen defects. Aryal et al. (2014) found that high expression of miRNAs on male flowers was related to IAA signaling in the study of sex differentiation of Carica papaya. However, ARF, SAUR, PYL, PLR, and other genes were significantly expressed in the signaling pathway of IAA and ABA in the male and female flower buds of J. mandshurica, indicating that these two plant hormones could regulate the differentiation of male and female flower buds. The endogenous GA3, ZT and ABA of loquat (Eriobotrya japonica) flower were determined during its development, and it was found that there was a high correlation between the change of endogenous hormone concentration and the expression level of genes related to hormone signal transduction pathway (Jing et al., 2020). In addition, transcriptome analysis of Madhuca pasquieri showed that auxin, GA, ABA and cytokinin metabolism pathway were related to seed germination and post germination (Kan et al., 2020). ETH also plays an important role in the gender differentiation of plants. Duan et al. (2008) found that the increase in ETH content affected the development of stamens during the flower bud differentiation of A. thaliana, and stamens were most sensitive to ETH in all flower organs. ERF was downregulated in the ETH signaling pathway of male flower buds of J. mandshurica. The expression levels in female flower buds and male flower buds are different. In the determination results of ETH, we found that the ETH content was at a low level before April 2, and began to increase greatly after April 2, which may indicate that ETH played an important role in the physiological differentiation of flower buds. Additionally, the change trend of ABA content is similar to that of ETH, which may have synergistic effect. The most important physiological function of CTK is to induce bud differentiation, its specific role in flower bud differentiation is not clear, but it is known that CTK is involved in regulating the cell division and differentiation of flower meristems (Jacqmard et al., 2002). Nibau et al. (2011) found that SUP, a regulatory gene for flower organ development, controls the development of male and female flowers by acting on CTK signal transduction. In the process of flower bud differentiation of J. mandshurica, A-ARR was downregulated in the CTK signal transduction pathway of male flower bud, but there was no differential expression in a female flower bud, indicating that A-ARR may be involved in the flower bud gender differentiation of J. mandshurica. Wang (2019) found that BR can participate in many processes of regulating plant growth and development, but also interacts with other hormones, interacting with the IAA signal pathway through ARF. However, the CYCD3 gene was upregulated in the BR signaling pathway in female flower buds of J. mandshurica, but it was not differentially expressed in male flower buds. SA is a negative regulator of FLC and other flowering inhibition genes in the autonomous flowering pathway (Wada et al., 2010), and the PR-1 gene in the SA signaling pathway in the flower buds of J. mandshurica is downegulated, which may also have a certain influence on the flowering of J. mandshurica. TFs play an important role in the reproductive development of J. mandshurica. Studies have demonstrated that TFs play an important role in the sex determination and differentiation of plants and animals. In our study, many TF genes were expressed differently in the male and female flower buds of J. mandshurica. In different TF families, bHLH, MADS, NAC, MYB, and ARF are significantly enriched, indicating that they play an important role in the sex determination or differentiation of J. mandshurica. Members of these families are involved in the gender differentiation of other monoecious or monoecious plants (Akagi, Henry & Comai, 2014; Martin et al., 2009). By sequencing the transcripts of male and female flower buds of J. mandshurica, we found a large number of candidate differential genes involved in various biological processes, which may be related to the regulation of the formation of male and female flower buds. Combined with the determination results of endogenous hormone content in male and female flower buds at the early stage of physiological differentiation, it demonstrates that the change in hormone content does affect the flowering time, and the contents of IAA, BR, and SA generally demonstrate an upward trend during the development of flower buds. This indicates that there was a significant correlation between endogenous hormone levels in plants and gender differentiation, which was consistent with the research results of Zhong & Qin (2010). Each plant hormone does not independently regulate the expression of plant sex, but a variety of hormones form a network regulation model to interact and influence the gender differentiation of plants. Metabolic substances and energy metabolism are very vigorous in the process of flower bud differentiation of J. mandshurica. Cell growth, proliferation, and division are very fast, and hormone signal transduction is involved in the process of gender differentiation. Simultaneously, it is influenced by many environmental factors and endogenous hormones, especially the signal transduction and synthesis of endogenous hormones such as IAA, GA, CTK, and SA, which are the main reasons for the emergence of two mating types of J. mandshurica in the process of flower bud sex differentiation. In order to further understand the specific biological process and gene regulation related to sex differentiation of J. mandshurica, we will determine the proteome and metabonomics of J. mandshurica flower buds and the hormone content in the leaves and fruits in the future research, and further explore the regulation mechanism of sexual differentiation in J. mandshurica.

Conclusions

In this study, it was found that participation in the sex expression of J. mandshurica was related to metabolic pathways such as IAA, GA, CTK, etc. Through analysis and verification, the signal transduction factors involved in plant hormone synthesis were screened. The downregulated genes in the different genes between male and female flower buds were generally higher than the upregulated genes, and many of them inhibited flowering. When the genes such as SAUR, DELLA, and A-ARR are downregulated, flowering is promoted. During the process of flower bud sex differentiation, hormones play a leading role in regulating the growth and differentiation of flower organs.

Supplemental Information

Supplemental Information 1 E-value distribution of BLAST hits for each unigene.

Click here for additional data file.

Supplemental Information 2 Similarity distribution of the top BLAST hits for each unigene.

Click here for additional data file.

Supplemental Information 3 Species distribution of the top BLAST hits.

Click here for additional data file.

Supplemental Information 4 GO function annotion and classification of Unigene for transcriptome in Juglans mandshurica.

Click here for additional data file.

Supplemental Information 5 KEGG classification of Unigene for transcriptome in J mandshurica.

Click here for additional data file.

Supplemental Information 6 Statistics of clean reads for transcriptome in Juglans mandshurica.

Click here for additional data file.

Supplemental Information 7 Raw data exported from the number of differentially expressed genes in male and female flower buds for Fig. 8.

Click here for additional data file.

Supplemental Information 8 Validation of 12 randomly selected differentially expressed genes (DEGs).

DEGs derived from RNA-seq using qRT-PCR. gene IDs: TRINITY_DN51855_c0_g2, TRINITY_DN46777_c1_g1, TRINITY_DN49183_c5_g1, TRINITY_DN46240_c7_g3, TRINITY_DN48811_c2_g3, TRINITY_DN50192_c3_g6, TRINITY_DN49124_c1_g5, TRINITY_DN51527_c0_g2, TRINITY_DN43548_c0_g1, TRINITY_DN51809_c1_g3, TRINITY_DN52421_c7_g3, TRINITY_DN54040_c1_g2

Click here for additional data file.

Supplemental Information 9 Raw data related to the validation of 12 randomly selected differentially expressed genes (DEGs) derived from RNA-seq using real-time quantitative RT-PCR.

Click here for additional data file.

Supplemental Information 10 Raw data related to the changes of endogenous hormones in buds of Juglans mandshurica Maxim.

Auxin IAA, Cytokinin CTK, Gibberellin GA, ethylene ETH, brassinoids BR, Abscisin ABA, Salicylic acid SA, jasmonic acid JA

Click here for additional data file.

Supplemental Information 11 Data related to differential expressed genes.

Click here for additional data file.

Supplemental Information 12 Data related to 56399 unigenes of total 138,138 unigenes.

Click here for additional data file.

Supplemental Information 13 Data related to all 81739 unigenes of total 138,138 unigenes.

Click here for additional data file.

We thank the Shanghai Personalbio Biotechnology Company for assisting with the sequencing analysis. We are grateful to the editor and reviewers for their insightful comments and suggestions.

Additional Information and Declarations

Competing Interests

Author Contributions

DNA Deposition

Data Availability

The authors declare that they have no competing interests.

Baiting Qin conceived and designed the experiments, performed the experiments, analyzed the data, prepared figures and/or tables, and approved the final draft.

Xiujun Lu conceived and designed the experiments, performed the experiments, analyzed the data, prepared figures and/or tables, authored or reviewed drafts of the paper, and approved the final draft.

Xiaomei Sun analyzed the data, authored or reviewed drafts of the paper, and approved the final draft.

Jianguo Cui conceived and designed the experiments, performed the experiments, authored or reviewed drafts of the paper, and approved the final draft.

Jifeng Deng analyzed the data, authored or reviewed drafts of the paper, and approved the final draft.

Lijie Zhang conceived and designed the experiments, performed the experiments, analyzed the data, prepared figures and/or tables, authored or reviewed drafts of the paper, and approved the final draft.

The following information was supplied regarding the deposition of DNA sequences:

The raw sequences of our study are available at the Sequence Read Archive (SRA) of National Center for Biotechnology Information (NCBI): PRJNA693587.

The following information was supplied regarding data availability:

Individual raw data are described in the Supplemental Files.

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
