# Peer review of "Transcriptome-based analysis of the hormone regulation mechanism of gender differentiation in Juglans mandshurica Maxim"

_PeerJ, doi:10.7717/peerj.12328_

## Round 0.1 · original submission · Major Revisions

Dear Dr. Zhang and co-authors,

As you will see, all the reviewers found that your work was interesting.
However, they provided several comments and suggestions to you.

Reviewer 1 had several concerns about the manuscript organization and language styles. Reviewer 2 provided some important suggestions. For example, you should add more information about the gene expressions relevant to the hormone detection. This reviewer also suggests that you open your files containing all expression data and functional annotation.

I also think that your figure presentation/own conclusion can be improved. Reviewer 3 felt that this manuscript was well-written and had only minor comments.

You should consider them to strengthen your manuscript. I would like to ask you to address or to respond with reasons not to follow the
the suggestion made by these reviewers.

Finally, the authors should have the manuscript edited by a fluent English speaker.

Best regards,
Atsushi Fukushima

Reviewer 1 ·

Basic reporting

The language of this manuscript is crude.

Experimental design

No comment.

Validity of the findings

No comment

Additional comments

This commentary performed transcriptome sequencing of female and male flower buds at different development stages. And found that hormone signal transduction pathways play an important role in the sex differentiation of J. mandshurica. Then the gene expression patterns were verified by q-PCR, and endogenous hormones were determinated.
Overall, the idea of the manuscript is clear. However, some issues still need to be improved:
1. The language of this manuscript is crude, some descriptions are difficult to be read, and an embellishment on the language is required.
2. In lines 39-43, first you said dioecious mating system is not common in flowering plants, but then you said only only Batulaceae, Corylus, Juglandaceae, Carya, Juglans, and Cyclocarya are woody plants with a monoecious mating system. The sentence reads contradictory. Please recheck this sentence to make it clear.
3. All the text in the manuscript need justify align. A space is required before the reference in the manuscript in lines 38, 40, 44 et al.
4. Figures 2 displayed the original data, which should be removed to supplementary materials. Figures 3,4, 5 are too simple and could be integrated into one figure. The contents of figures 7 and 7 are similar and thus should also be integrated into one figure.
5. The quality of Figure 9, 10 is poor, please provide the original figure.
6. Figure 9, the RNA seq data also need an error bar.

Reviewer 2 ·

Basic reporting

This paper is general clearly describe and easy to understand. But there still some sentences need to be rewrtie or modified. Such as: 1.In the introdcution part line 70 to line 74. I was confused by this sentence. 2. In the abstract, line 29-line 30, ARF and SAUR were expressed differently in the auxin signaling pathway. I don't understand this sentence.

Experimental design

This experimental design is well defined and meaningful.

Validity of the findings

More data details need to be added. And the conclusions need to be developed.

Additional comments

This paper is well designed and the result seems to be very intresting. I beleived this paper could be improved a lot. I suggest to add more information about the hormone synthesis genes expression relevant to the hormone detection. Meanwhile ,some figures need to be improved. Such as the heatmaps. The hormone detection result is very important, this should be discussed in the discussion part and should be mentioned in the abstract. The sample collection should be described more clearly, since it is very improtant. The file containing all genes' expression data, annotation and symbols should be provied.

Annotated reviews are not available for download in order to protect the identity of reviewers who chose to remain anonymous.

Reviewer 3 ·

Basic reporting

Authors used male and female flower buds of J. mandshurica at the physiological differentiation stage. Illumina-based, transcriptome sequencing was performed, and the quality of the sequencing results was evaluated and analyzed. Findings are not extremely novel, but they do perfect fill a regional knowledge gap.

Experimental design

no comment

Validity of the findings

The introduction is well-written and concise. The methods are exceptionally clear compared to most manuscripts I've reviewed on the subject. Results are well constrained and the author does not over-extend the discussion or conclusions. I do not often compliment manuscripts in review, but this is one of the most straightforward manuscripts I have read in a very long time and that was refreshing.

Additional comments

Before the paper is accepted, there are several questions should be considered:
1. The details of geography distribution of J. mandshurica should be mentioned.
2. The future work in Discussion section need to be added.
3. The Conclusion section should be enriched, and the last sentence need to be deleted.

---

## Round 0.2 · Minor Revisions

Dear Dr. Zhang and co-authors,


In this review round, our reviewers found that you have answered and responded well. However, the Section Editor Dr. Gerard Lazo and I had some comments and suggestions to improve the current version of the manuscript. The important points are as follows:

(1) First, we would recommend professional copy editing and review,
because many of the sentences/phrases are incomplete and unclear.

(2) To make a biologically meaningful discussion about differentially expressed genes in male and female flower buds, the authors should describe the details of gene ontology (GO)- and pathway-based functional analysis. For example, the GO annotation analysis needs to concrete associations between assembled contigs and their associated identifications. The statistics is also important:

- the number of genes in your cluster annotated to a certain GO category
- the total number of genes in your cluster.
- the number of genes in the reference set etc...

(3) There are some *.rar files included; however, it may be best presented in another format such a *.zip, *.tar.gz, or even *.bz2, but to have the content meaningfully organized so distinguish sequences and annotation linkages. If there is a data repository of preference (NCBI, figshare, or others) that may even suffice as long as the data can be connected with what is presented in the manuscript. I think that the authors can deposit your assembled unigenes into TSA
(https://www.ncbi.nlm.nih.gov/genbank/tsa/) before publication.


Best regards,
Atsushi Fukushima

Reviewer 1 ·

Basic reporting

no comment

Experimental design

no comment

Validity of the findings

no comment

Reviewer 2 ·

Basic reporting

no comment

Experimental design

no comment

Validity of the findings

no comment

Additional comments

The authors addressed the questions well.

For Figure 11, the statistical analysis needs to be added. Or change the figure into a table containing
all the information

Reviewer 3 ·

Basic reporting

No comment

Experimental design

No comment

Validity of the findings

No comment

Additional comments

No comment

---

## Round 0.3 · Minor Revisions

Dear authors,

Thank you for revising. I still have a question and a suggestion.

(1) Where is the detailed description of gene ontology (GO) and
functional analysis? As a whole, your responses are too simple.

(2) Would you please deposit all the assembled unigenes into NCBI TSA
(https://www.ncbi.nlm.nih.gov/genbank/tsa/)? After that, please add
your TSA's accession number in the revised manuscript. I think this is for improving the reusability.

Best regards,
Atsushi Fukushima

---

## Round 0.4 · Minor Revisions

Dear authors,

Would you please revise your manuscript according to the comments provided by our Section Editor?

"Again, “The images representing the GO term spread is not enough; there need to be concrete associations between assembled contigs and their associated identifications.”; this was not addressed.

Again, “There are some *.rar files included; however, it may be best presented in another format such a *.zip, *.tar.gz, or even *.bz2, but to have the content meaningfully organized so distinguish sequences and annotation linkages.” Please DO NOT use *.rar formatted data. I have unrar installed and was not able to cleanly extract data, and there are free and non-free versions; a more open-sourced packing software would be more appropriate. Or again, try again to deposit data in a third-party resource that would accept the data being generated as mentioned earlier.

I still recommend further revision is needed at this point. Very little was addressed from the previous decsion."

Best regards

---

## Round 0.5 · Minor Revisions

Dear authors,

Thank you for revising. However, our section editor has commented as follows:

"I can now see the assembled TRINITY sequences from the zip data files provided; however I still think it would be better placed in a third-party data repository rather than here (however this may do).

It was good to add the GO: annotations with respect to the annotation classification, but there is still no connection between the annotation and the sequence (it is just summary data). If a figure is represented it is important to know how the sequences presented matched up to the reported data. Especially since data is pooled in a Venn diagram as shown in Figure 8; which sequences are in each of the 15 classifications. Just the same as seeing the histograms provided, which sequences are in each of the columns.

Such explanations can be adjusted by providing a table further describing classifications, libraries for each of the assembled sequences to support your summary illustrations.

The manuscript is still in need for further clarification. Attention needs to realize how the reader will interpret what is presented. There was an improvement in clarifying some items, but more is still required."

Would you please improve your manuscript?

Best regards

---

## Round 0.6 · Minor Revisions

Dear authors,

First, I would suggest that you use the TSA in NCBI as follows.

Transcriptome Shotgun Assembly Sequence Database
https://www.ncbi.nlm.nih.gov/genbank/tsa/

The second choise may be to use the FigShare. For example, see
https://figshare.com/articles/dataset/Trinity_assembled_contig_files_used_for_Phyluce/11380371/1.

Best regards,
Atsu

---

## Round 0.7 · Minor Revisions

Dear authors,

Thank you for uploading your assembled transcripts into TSA. However, you should carefully revise the manuscript according to the following comments.

The Section Editor, has commented and said:

"The rebuttal letter indicates that SUB10193123 was submitted; however, there is no mention of it in the revision sent. The description of how to connect to the data should be placed within the context of the manuscript. I did see some revisions placed in the manuscript, but none pointed to the resources previously requested. The main subject in the abstracts focuses on 8116 sequences; up- and down-regulated. The FASTA transcripts from the de novo assembly need to be made available from a resource with appropriate annotations connecting them to the analysis stated within the manuscript, the Venn diagram in Figure 8 nor the histograms do anything to make these connections, except to summarize observations. There needs to be serious consideration of how to make these connections; the manuscript is still considered in need of revision."

I would like to give a chance to improve your manuscript thoroughly. If you do not address the comments above, including the openness of your data, I will have to reject it next time.


Best regards

---

## Round 0.8 · accepted · Accept

Dear authors,

Thank you for revising.

Best regards